# Airway Epithelium: A Neglected but Crucial Cell Type in Asthma Pathobiology

**DOI:** 10.3390/diagnostics13040808

**Published:** 2023-02-20

**Authors:** Sabita Singh, Joytri Dutta, Archita Ray, Atmaja Karmakar, Ulaganathan Mabalirajan

**Affiliations:** 1Molecular Pathobiology of Respiratory Diseases, Cell Biology and Physiology Division, Council of Scientific and Industrial Research (CSIR)-Indian Institute of Chemical Biology (IICB), Kolkata 700091, West Bengal, India; 2Academy of Scientific and Innovative Research (AcSIR), Sector-19, Kamla Nehru Nagar, Ghaziabad 201002, Uttar Pradesh, India

**Keywords:** allergic asthma, airway epithelium, alarmins

## Abstract

The features of allergic asthma are believed to be mediated mostly through the Th2 immune response. In this Th2-dominant concept, the airway epithelium is presented as the helpless victim of Th2 cytokines. However, this Th2-dominant concept is inadequate to fill some of the vital knowledge gaps in asthma pathogenesis, like the poor correlation between airway inflammation and airway remodeling and severe asthma endotypes, including Th2-low asthma, therapy resistance, etc. Since the discovery of type 2 innate lymphoid cells in 2010, asthma researchers started believing in that the airway epithelium played a crucial role, as alarmins, which are the inducers of ILC2, are almost exclusively secreted by the airway epithelium. This underscores the eminence of airway epithelium in asthma pathogenesis. However, the airway epithelium has a bipartite functionality in sustaining healthy lung homeostasis and asthmatic lungs. On the one hand, the airway epithelium maintains lung homeostasis against environmental irritants/pollutants with the aid of its various armamentaria, including its chemosensory apparatus and detoxification system. Alternatively, it induces an ILC2-mediated type 2 immune response through alarmins to amplify the inflammatory response. However, the available evidence indicates that restoring epithelial health may attenuate asthmatic features. Thus, we conjecture that an epithelium-driven concept in asthma pathogenesis could fill most of the gaps in current asthma knowledge, and the incorporation of epithelial-protective agents to enhance the robustness of the epithelial barrier and the combative capacity of the airway epithelium against exogenous irritants/allergens may mitigate asthma incidence and severity, resulting in better asthma control.

## 1. Introduction

The Global Strategy for Asthma Management and Prevention (2022 update) describes asthma as a ‘heterogeneous’ disease, usually characterized by chronic airway inflammation, and it is defined by the history of respiratory symptoms, such as wheeze, shortness of breath, chest tightness, and cough that vary over time and in intensity, together with variable expiratory airflow limitations. Airflow limitation may later become persistent [1]. Asthma affected an estimated 262 million people in 2019 and caused 455,000 deaths globally [2]. The two major types of asthma, atopic and allergic asthma, have an early childhood onset and are typically linked with a family history of asthma and/or other diseases of the atopic triad, namely allergic rhinitis and atopic dermatitis. In atopic asthma, various types of allergens, like different types of common pollens, cockroach allergens, house dust mites, dander from cats and dogs, etc., are the major inducers. These allergens sensitize only those individuals who are allergy-prone and have a genetic tendency [3]. After sensitization, when these allergy-prone individuals are exposed to the same allergens again, it leads to bronchoconstriction. Various forms of nonatopic asthma also exist, which include occupational asthma and exercise-induced asthma. Occupational asthma is caused by inhaling fumes, gases, and dust particles at the workplace. It is generally a reversible condition, wherein the symptoms disappear on aversion from the causative irritants. In exercise-induced asthma, the respiratory rate and tidal volume are heightened during exercise, which requires the airways to humidify a large amount of air in a short period, which causes excessive water loss from the airways. Consequently, hyperosmolarity and relative dehydration occur, which triggers the release of mediators like prostaglandins, histamine, leukotrienes, and cytokines that causes airway smooth muscle contraction and airway narrowing, leading to asthma [4].

Based on quantitative sputum cell analysis, Simpson and coworkers classified airway inflammation involved in asthma into eosinophilic, neutrophilic, paucigranulocytic, and mixed granulocytic types. The neutrophilic subtype is triggered by lung irritants like cigarette smoke, air pollutants, inhaled ozone, cold environment, vigorous exercise, or infectious exacerbation and is generally refractory to corticosteroid response. Mixed granulocytic asthma is characterized by the coexistence of eosinophils and neutrophils and might represent a transitional phase in asthma progression. Paucigranulocytic airway inflammation entails airway hyperresponsiveness due to altered airway smooth muscle contractility without the presence of eosinophils or neutrophils in the airways [5].

Nevertheless, allergic eosinophilic asthma is the most common type of asthma. The pathophysiology of asthma features airway obstruction accompanied by a narrowing of the airway lumen diameter owing to the chronic inflammation of the airways in response to inhaled allergens. There is plasma extravasation, edema, and an influx of various immune cells like eosinophils, neutrophils, lymphocytes, macrophages, and mast cells. Airway hyperresponsiveness occurs, wherein the smooth muscle of the airways contracts in response to inhaled stimuli, which can be reversed by the administration of bronchodilator. Apart from this, mucus plugging, airway remodeling, which includes goblet cell metaplasia, excessive subepithelial collagen deposition, airway smooth muscle hyperplasia, and increased vascularity are the possible mechanisms of this persistent airflow obstruction [6].

In allergic asthma, the exposure of sensitized individuals to allergens evokes a type 2 immune response. During the sensitization phase, dendritic cells capture inhaled allergens and present them to naive CD4+ T cells that are polarized to T helper 2 (Th2) cells, which secrete cytokines like IL-4, IL-5, and IL-13. IL-4 directs immunoglobulin class switches to IgE in B cells, which binds to the high-affinity IgE receptor on mast cells. Allergen re-exposure prompts allergen-mediated IgE crosslinking that results in rapid mast cell activation and degranulation. IL-5 brings about airway eosinophilia. IL-4 and IL-13 exert their direct effect on the airway epithelium by inducing goblet cell metaplasia and mucus hypersecretion. Moreover, IL-13 elicits airway hyperresponsiveness by acting on the airway smooth muscle cells [7]. Thus, the traditional concept of asthma involves the Th2-dominant immune response. However, these Th2 cytokines are also produced by type 2 innate lymphoid cells (ILC2s) to orchestrate the Th2 immune pathway in asthma. Importantly, the inducers of ILC2 are cytokines that are almost exclusively secreted by the airway epithelium. This underscores the importance of airway epithelium in asthma pathogenesis. 

## 2. Need to Change the Existing Concepts of Asthma Pathogenesis (Figure 1)

A variety of medications are available in asthma therapy, and among them are bronchodilators and anti-inflammatory drugs, which include immunosuppressive agents. These currently available medications are based on the changes in the conceptual understanding of asthma pathogenesis since the era of Hippocrates. However, the obvious question lies in the exigency of novel concepts in the manifestation of asthma pathogenesis. While the current Th2-dominant concept of asthma not only failed to explain the dissociation between airway inflammation and airway remodeling, it also failed to explain the reasons for the increased incidences of severe asthma and therapy-resistant asthma. All these indicate a need for newer concepts in asthma pathogenesis that can explain the above pathological features on the basis of which one can devise novel therapeutic strategies for severe asthma and therapy-resistant asthma. In fact, the currently available medications are also based on the change in earlier concepts of asthma pathogenesis. 

**Figure 1 diagnostics-13-00808-f001:**
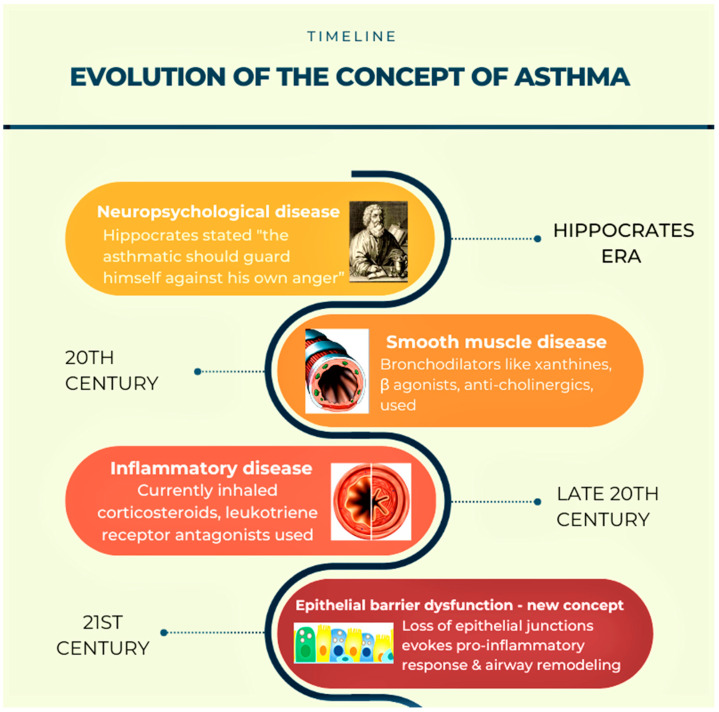
A diagram that describes the concept changes that happened in asthma pathogenesis with time.

The first description of asthma was propounded by the Greek physician Hippocrates (460–377 BC) and is derived from the Greek word ‘‘asthmaino’’ (astZmaino), i.e., ‘‘panting or gasping’’. Early regimens to treat asthma were focused on symptom relief or the alteration of external factors, such as the use of plant extracts, life-style changes, surgery, or hypnosis. In the pre-inhaler era, inhaling the smoke from ‘‘asthma cigarettes’’, which contained compounds like atropine, belladonna, menthol, morphine, or cocaine, was also prevalent. For many years, the concept of the neuro-psychogenic origin of asthma was widely accepted. Due to this, psychopharmaca, like chlorpromazine, was prescribed. In 1892, the father of modern medicine in the Western World, Sir William Osler described asthma as the spasm of the bronchial muscles. As a result, bronchodilators like epinephrine, anticholinergics, and inhaled β-agonists were all introduced in the first half of the 20th century. The concept of asthma being an inflammatory disorder was strongly established in the 20th century. Therefore, targeting inflammatory mechanisms became popular in the second half of the 20th century. Presently, corticosteroids are hailed as the most potent anti-inflammatory drugs for asthma therapy. The action of leukotriene receptor antagonists is based on the antagonism of cysteinyl leukotrienes (CysLTs) at the CysLT1-receptor in the airways and inflammatory cells, which mediate bronchoconstriction, inflammation, and mucus production in asthma. Although they are not as efficacious as corticosteroids, they are used as an add-on medication [8,9,10].

Although the current therapeutic approach to asthma focuses mainly on the resolution of inflammation, it fails to ameliorate lung function or address disease exacerbation, which indicates the involvement of other factors. An upcoming concept projects airway remodeling as the other major culprit in asthma pathogenesis [11]. The disruption of the tight epithelial junctions in the airways confers susceptibility to allergens and evokes inflammatory responses, contributing to airway remodeling. In the airway epithelium of children with respiratory difficulties, structural alterations have been observed prior to the advent of airway inflammation and the clinical detection of asthma. This supports the notion that epithelial changes are an early event in asthma pathogenesis, thus challenging the dogma that chronic airway inflammation begets airway remodeling [12]. This concept of epithelial dominancy in asthma pathogenesis is relatively new in the context of the existing Th2-dominant concept. There is a need for new concepts in the context of therapy resistance and increased morbidity in asthma. Though it is well-known that epithelial injury and its subsequent changes, like airway remodeling, are crucial in asthma pathogenesis [13], detailed mechanistic studies were missing that could explain airway epithelial injury. Initially, it was believed that airway epithelial injury and airway remodeling were mere consequences of airway inflammation. However, Th2-dominant airway inflammation alone could not explain all these complex asthma features [14].

Thus, it is high time to acknowledge the pivotal role of the airway epithelium in driving the pathogenesis of asthma and delve deeper into understanding its mechanistic role.

## 3. Barrier Function of Airway Epithelium 

### 3.1. Importance of Epithelial Barrier in Maintaining Homeostasis

As a first line of nonspecific defense, the anatomical barriers such as, skin and mucosal membrane protects our body system from environmental insults. Essentially, these barriers in our body have two major functions: (a) organ-specific functions and (b) the maintenance of organ homeostasis against external aggressors. For example, the stratum corneum, which is the outermost epidermal layer, is essential in limiting water loss by transcutaneous evaporation to maintain the water content of our body [15]. In addition, the skin barrier is also crucial in providing defense against the invasion of various external molecules and microbes. Similarly, the intestinal epithelium is essential in the absorption of nutrients, water, and electrolytes, along with the homeostatic role of restraining the entry of allergens, microbes, and other foreign molecules into the intestinal wall. Likewise, alveolar epithelium is fundamentally involved in gaseous exchanges, surfactant production, and the regulation of ions and water transport to maintain the fluid balance on the alveolar surface, in addition to the protective function against environmental irritants/pollutants [16]. The airway epithelial cells, which line the entire region from the trachea to the terminal bronchi, are lined up by the ciliated cells and the Clara cells at the region of conducting airways (trachea, bronchi, and bronchioles). However, airway epithelium does not have any special organ-specific function like gas exchange or nutrient absorption, even though it regulates water/ions transport. However, airway epithelial cells serve as sentries in restricting the entry of inspired airway luminal contents beneath the epithelial layer and in removing/neutralizing the various toxic/irritant substances from the inhaled air to prevent the access of these irritants to alveoli where vital gas exchange happens. In order to perform this key function, the airway epithelial layer has numerous types of machinery, like physical barriers, chemical barriers, special cellular machinery, etc.

Meanwhile, the regulation of the airway epithelial barrier function is emerging as a crucial checkpoint in asthma pathobiology. Numerous studies have evidenced that the respiratory epithelium is compromised in asthmatic conditions. Additionally, it has been noted that asthma patients’ bronchial epithelial cells have abnormal antimicrobial response patterns. According to biopsy studies carried out in children, structural alterations in the respiratory epithelium may take place prior to the beginning of airway inflammation. There is a theory that claims that structural and functional dysfunction in the respiratory epithelium leads to an aberrant immune system and structural cell signaling, which, in turn, causes remodeling, inflammation, and allergic airway hypersensitivity [17].

### 3.2. Anatomical Barrier Role of Airway Epithelium (Figure 2)

Epithelial junctional complexes act as crucial signaling hubs and threat detectors, interacting with the microenvironment and monitoring self-defense. This function of the epithelial barrier of the bronchial epithelial cells and its structural integrity is mainly conferred by the adhesive forces of the intercellular junctions. The major intercellular junction proteins documented in preserving the barrier include tight junctions, the adherens junctions, and the (hemi) desmosomes [18]. The tight and adherens junctions together form the apical junctional complex (AJC) and are present at the apicolateral border of the airway epithelial cells.

**Figure 2 diagnostics-13-00808-f002:**
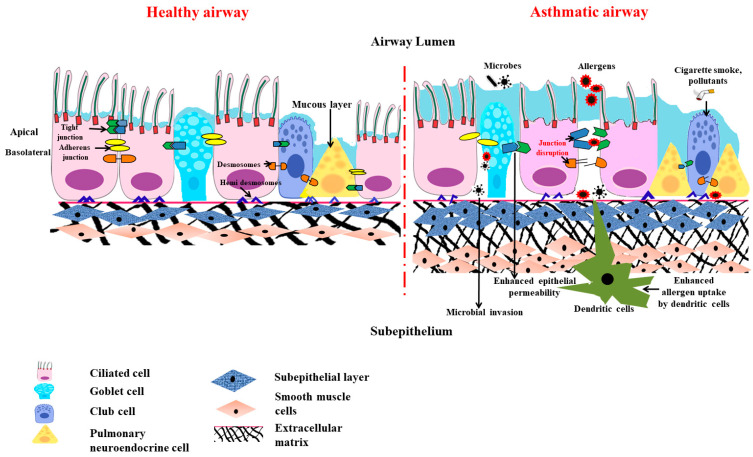
Scheme diagram to illustrate the structural changes in the airway epithelial barrier in asthmatics compared to healthy airway epithelium. When the asthmatic airway epithelium is exposed to allergens and air pollutants, it causes disruption to the tight epithelial junction and adherens junction. This leads to heightened mucosal permeability, effectuating more inhaled particles and allergens present in the subepithelial region and promoting innate and adaptive immune responses. This is accompanied by PNEC hyperplasia, a loss of ciliated cell numbers, goblet cell metaplasia, mucus hypersecretion, the thickening of the basal membrane, subepithelial fibrosis, increased airway smooth muscle mass, and the excess deposition of extracellular matrix.

The claudin multigene family, which encodes for the tetraspan transmembrane proteins, are the core tight junction proteins [19]. They interact with the claudins on the adjacent cells and forge a barrier that regulates the paracellular diffusion of ions and solutes. The alveolar epithelium also expresses over a dozen claudins, out of which claudin-3 (cldn-3), claudin-4 (cldn-4), and claudin-18(cldn-18) are predominantly expressed [20]. The other claudins expressed by the alveolar epithelia include claudin-5 (cldn-5) and claudin-7 (cldn-7), which also aids in maintaining the alveolar epithelial barrier. The loss of these proteins leads to increased permeability and causes a loss of barrier function [21]. A research study conducted on the tight junction proteins involved in airway epithelial cells demonstrated that the expression of claudin-18 had been reduced in patients with asthma [22]. Asthmatic mice with claudin-18 deficiency were found to manifest increased susceptibility to airway hypersensitivity, which indicated the contribution of claudin-18 to the pathophysiology of asthma [22]. 

The adherens junctions that are present beneath the tight junction comprise the cadherin (E-cadherin) and the catenin (α-catenin, β-catenin, and p120) families that mediate cell-to-cell adhesion [23,24]. The deletion of E-cadherin in lung epithelia not only causes epithelial denudation with a loss of ciliated cells but also suppresses the differentiation of club cells and, thus, limits epithelial healing after damage due to the stem cell properties of club cells [25]. The sustained loss of E-cadherin also evokes the differentiation of epithelia into a mesenchymal phenotype, a process otherwise called epithelial-mesenchymal transition (EMT). This EMT is known to cause subepithelial fibrosis, which is the main feature in airway remodeling in asthmatics and increases the severity and complexity of the disease. 

The tight and the adherens junction proteins are linked to cytosolic proteins on one end and to the actin cytoskeleton on the other end to form “cytosolic plaques” [26]. A dominant plaque protein is found in the family of zonula occludens (ZO), which cement the intracellular domains of the tight and adherens junction with various cytoskeletal components and actin-binding proteins, like α-catenin, α-actinin, and vinculin. It is now becoming evident that a lack of these certain junctional complexes is sufficient to trigger the alarm of the immune defenses. Perez-Moreno et al. demonstrated that the conditional ablation of p120 in the epidermal region provoked a robust inflammatory response [27]. The loss of p120 binding results in the nuclear translocation of nuclear factor kappa, which is a light chain enhancer of activated B cells (NF-κB), and thus drives the action of the inflammatory response via the promotion of IκBα [28]. Similarly, p120 downregulation in the human bronchial epithelial cells induces the activation of NF-κB and elicits the enhanced expression of proinflammatory cytokines, like IL-8 (partially mediated via the RhoA activity). There is growing evidence that asthmatic tissues more heavily activate NF-κB pathways. In samples from asthmatic patients, the inducible and transcriptionally active variants of NF-κB were shown, which corresponds to the increased expression of specific proinflammatory cytokines and enzymes observed in these patients [29]. 

Desmosomes and hemi-desmosomes connect the columnar epithelial cells to the basal cells and basement basal cells to extracellular matrix in the airways. They regulate cell adhesions and give mechanical backing to lung tissue [30]. 

### 3.3. Chemical Barrier Role of Airway Epithelium 

Given the incessant exposure of the lungs to noxious substances, the surface of the airway is layered up with a highly evolved fluid lining that is exchanged dynamically to aid in the mucociliary clearance of antigens. The produced mucus layer and the periciliary layer (PCL) are the two separate layers that make up this fluid lining, also known as airway surface liquid (ASL). The secreted mucins are glycoconjugates with threonine-rich domains, and these act as biophysical “rafts” or barriers to convey the pathogens out of the conducting airways [31]. Mucins, like MUC13, MUC16, and MUC4, which are tethered to the epithelial cells, bestow a direct host defense barrier at the epithelial surface that can be removed by pathogen- or host-associated proteases, releasing microbes to the mucociliary ‘escalator’ for elimination. On the contrary, MUC5AC, MUC5BA, and MUC2, which are secreted by the airways, create a mucous gel that hinders bacterial aggregation, binds microbial pathogens, and impairs their ability to adhere [31]. Even though the increased secretion and release of airway mucus is a noisome accompaniment to environmental cues and infections, mucins help maintain airway homeostasis and eliminate pathogens and cellular debris during recovery from injury or infection. Airway disorders viz., chronic obstructive pulmonary disease (COPD), bronchiectasis, cystic fibrosis, and asthma, all exhibit excessive goblet cell differentiation and mucus hyperproduction [32].

The epithelial cells and the submucosal glands, which form an integral part of the innate immune system, secrete many host-defense proteins, including lysozymes, lactoferrin, lipocalin-2, defensins, cathelicidin, surfactant proteins, acute-phase proteins, psoriasin (S100A7) proteins, and palate, lung, and nasal epithelium clone (PLUNC) proteins [33]. The lung epithelial cells act as immune sentinels, and this sets up an innate immune response owing to an ample repertoire of cytosolic, membrane-bound, and endosomal pattern-recognition receptors (PRRs) to identify various pathogens [34]. 

### 3.4. Physiological Barrier in Airway

The airway defense responses that guard the lungs and the rest of the body against inhaled irritants like cigarette smoke and aerosols include bronchoconstriction, which is a crucial and effective reflex mechanism [35]. However, the chemical irritants stimulate the sensory nerves present in the respiratory tract, and the generated action potential is conducted by the vagus nerves (to the brain stem). This instantly causes bronchoconstriction via the cholinergic efferent pathway accompanied by the hypersecretion of mucus, coughing, and dyspneic sensations [36]. Thereof, the conjecture that bronchoconstriction is a physiological protective mechanism, becomes a clinical symptom as this also confines the entry of the air and, thus, difficulty in breathing occurs subsequent to the bronchoconstriction. 

### 3.5. Special Cellular Machinery in Airway Epithelial Barrier

Surprisingly, airway epithelial layers also have special machinery, like pulmonary neuroendocrine cells (PNECs), that produce an array of neuropeptides, neurotransmitters, and amines to sense the environmental air and catabolize the irritants/toxicants of the air to neutralize them [37,38]. Therefore, these cells are considered intrapulmonary sensors that also sense hypoxia by chemoreception [39]. Indeed, like the traditional chemoreceptors present in the nose, the PNEC clusters that are part of the epithelial layer have been shown to have olfactory receptors so that they can sense the toxicants in the inspired air and react to prevent the further entry of such toxicants through induction of bronchoconstriction [40]. Recent research by Sui et al. has provided mechanistic proof that neuroendocrine cells play a crucial role in orchestrating asthmatic responses by activating goblet cells and group 2 innate lymphoid cells (ILC2). PNECs secrete calcitonin, gene-related peptide that activates the ILC2 response, which further encourages Th2 allergic reactions [41]. Studies have revealed an increase in PNECs in a variety of lung conditions, including asthma [42], COPD [35,39], and small-cell lung cancer [43].

Similarly, club cells present in the airway epithelial layer act like hepatocytes of the lung as they have more cytochrome P450 to detoxify xenobiotics present in the inspired air [44]. Club cells have the capacity to differentiate into ciliated and mucus-secreting goblet cells in response to epithelial damage, and this process is regulated by the intercellular junctional protein E-cadherin [45]. Patients with asthma and COPD [46] have been reported to have a reduced number of club cell counts, and this is associated with disease-severity [47].

In addition to having a chemosensory role, PNECs are also considered to be reserved stem cells, especially in the distal airway, and thus can form ciliated and Clara cells postepithelial injury [48]. Similarly, Clara or club cells are considered airway progenitor cells. A subset of variant Clara cells has been shown to have multipotent differentiation properties and, thus, can regenerate the bronchiolar epithelium [49]. 

### 3.6. Barrier Function of Airway Epithelium against Air Pollutants and Pathogens 

Air pollutants are typically categorized as ultrafine, fine, and coarse, depending on their size, source, and nature (gases or particles). The main sources of indoor air pollution include stoves, biological substances (including mold), microplastics, and household dust, whereas automobile, industrial (urban), and agricultural (rural) activity are the major reason for outdoor pollution [50]. Airborne particulate matter (PM) is a heterogeneous mixture of solids and aerosols, which can include heavy metals, airborne dust, and nanoparticles discharged from chemical factories, wildfire smoke, vehicle exhaust, and volcanic eruptions. Urban homes have PM_2.5_- and PM_10_-rich indoor pollutants along with higher NO_2_ levels [51]. Since fine PM penetrates the narrow airways more deeply than coarse PM, it is particularly dangerous to breathe it in. Inhalation of fine particulates has been linked to the development of asthma and COPD. A correlation has been put forward between increased levels of outdoor pollution, particularly NO_2_, PM_2.5_, and black carbon, and the onset and development of childhood asthma along with lowered lung function [52]. 

In general, air pollutants may not lead to the development of asthma in atopic individuals, but this promotes the initiation. For example, cigarette smoke has been shown to increase the accessibility of allergens by damaging the airway epithelial barrier [53]. As a result, air pollution could potentiate the allergen uptake followed by their processing by antigen-presenting cells. Thus, children residing in urban areas have an increased tendency to develop asthma than non-urban children [51] by virtue of more pollution. On the other hand, in non-atopic asthma, both indoor and outdoor pollutants could initiate asthma development and this leads to cause occupational asthma. Antigen-presenting cells (APCs) present PM entering the submucosa of the airway to adaptive immune cells. PM-mediated increase in MHCII on APCs expedites the proliferation of adaptive immune cells and the release of inflammatory cytokines that further perturbs the functioning of the airway epithelial barrier [54]. Additionally, exposure to PM increases the number of neutrophils in the submucosa, which may cause the release of surfactant proteins to cause barrier dysfunction [55]. All this evidence indicates that air pollution can aggravate asthma initiation and development through many mechanisms. The primary risk factors for chronic airway disorders are these environmental factors along with genetic susceptibility factors. Both atopic and occupational asthma involve sensitization, in one case via allergens and in the latter case through low-weight molecular agents. The prevalence of occupational asthma is similarly as high as atopic asthma. The kind of etiological agents or asthmagens influences the mechanism of occupational asthma. These asthmagens are conventionally categorized as low-molecular-weight (LMW) agents and high-molecular-weight (HMW), with the limit confined to 5 to 10kDa [56]. LMW-mediated occupational asthma is brought on by chemicals like diisocyanates (e.g., tolueen diisocyanate, TDI) [57], metals, and certain substances generated from wood. LMW sensitizers have hapten-like properties. The precise processes behind LMW-related OA have not yet been thoroughly defined, despite the fact that specific IgE antibodies have also been found in OA caused by several LMW agents and that multiple investigations have suggested that immunologic mechanisms are at play [58]. In addition to these factors, asthma and other allergic diseases tend to get aggravated with seasonal change. This is due to epithelial injury as the denuded airway epithelial layer leads to exposed nerve endings that are located beneath the epithelial layer to the environment directly. As a result, temperature variation in the environment directly irritates these bare nerve endings to cause bronchoconstriction via smooth muscle contraction. In addition to these physical factors, biological factors like more grass pollination in the spring season can also induce/aggravate asthma development [59]. 

In addition to pollutants and allergens, pathogens also have a major role in disrupting the epithelial barrier. Adivitiya et al. have shown the impact Sars Cov2 has on mucociliary clearance through protein network analysis. The spike (S) protein of the virus utilizes ACE2 (angiotensin-converting enzyme 2) and TMPRSS2 protease, which are present in the ciliated and secretory cells, to enter the host. The receptor-binding domain of S also interacts with CD209, a lectin protein found in the epithelial cell, to facilitate virus entry. Additionally, PPIA and Neurolipin1 host receptors play a major role in potentiating infection. The virus causes loss of cilia, which in turn causes reduced mucus secretion and pathogen clearance. This again highlights the importance of the epithelial barrier and mucociliary clearance in combating viral infections [32]. 

Overall, airway epithelium has a variety of resources to protect lung function against environmental irritants. 

## 4. The Victim Role of Airway Epithelium in Asthma Pathogenesis 

From the earlier section, we can come to the clear conclusion that airway epithelium is not just part of the conducting airway, but it acts beyond the mere role of a physical barrier. However, in an immune era, its exact role in lung diseases, including asthma, has been underestimated. 

### 4.1. Role of Th2 Cytokines in Airway Epithelial Barrier Dysfunction

It is generally believed that T helper 2 lymphocytes play a crucial role in asthma development after the initial sensitization phase (with allergen exposure). Upon repeated secondary exposures to the same allergen, Th2 cells accumulate in the lungs, and the features of allergic asthma develop. This is popularly referred to as type2/Th2 asthma, whereby such Th2 cells release various cytokines, like IL-4, IL-5, and IL-13, in response to allergens [60]. 

**(a) IL-4 and IL-13:** Both IL4 and IL-13 are crucial Th2 cytokines responsible for immunoglobulin class switching to enhance IgE production [61], leading to the degranulation of mast cells and basophils. The released proinflammatory meditators not only caused bronchoconstriction, but also cause airway epithelial injury. Treatment with both IL-4 and IL-13 cytokines showed reduced expression in the apical junctional complex proteins that encompass both the tight junction and adherent junction [62]. Treatment with IL-4 in HBEC (human bronchial epithelial cells) increases permeability in epithelial cells. There is reduced transepithelial electrical resistance that, in turn, leads to increased allergen sensitization and allergen uptake. IL-4 and IL-13 also cause a reduction in the ciliary beating of the ciliated epithelial cells and, in turn, in mucociliary clearance [45]. IL-13 also plays a major role in the secretion of periostin (POSTN), which is an essential biomarker in asthma. Periostin is an extracellular protein present in the matrix. It is a downstream product of the IL-13 pathway, signifying type 2 immunity. POSTN release is then coupled with epithelial mesenchyme transition, which is also shown in vitro in Beas2B cell lines. Thus, POSTN plays a major role in airway remodeling [63]. Overall, these cytokines are responsible for injuries to the airway epithelium. In addition to causing airway epithelial injury through oxidative stress, both IL-4 and IL-13 have been shown to cause perturbation in airway epithelial integrity with barrier dysfunction [62,64]. 

**(b) IL-5:** IL-5 plays a crucial role in differentiation, activation, maturation, and the recruitment of eosinophil that releases major basic cationic proteins, like major basic protein-1 (MBP-1), eosinophil cationic protein (ECP), eosinophil derived neurotoxin (EDN), and eosinophil peroxidase (EPO-1), that induce oxidative stress, causing an injury to the airway epithelium [65].

### 4.2. Mitochondrial Dysfunction in Asthmatic Airway Epithelium

The above section demonstrated how Th2 cytokines could cause a loss of epithelial barrier function. This section will focus on how these cytokines are linked to mitochondrial dysfunction in asthmatic airway epithelium.

The earlier clinical study found the presence of an increased number of mitochondria with altered structures in asthmatic children [66]. Later, our lab demonstrated the involvement of mitochondrial dysfunction in asthma pathogenesis [67]. Our lab also demonstrated a reduction in the cytochrome c oxidase (COX), which is the primary enzyme of the electron transport chain (ETC) residing in mitochondria’s inner mitochondrial membrane (IMM), which transfers electrons from the cytochrome c to oxygen in the lungs of mice with allergic airway inflammation. Further, the key subunit of cytochrome c oxidase was found to be reduced in the bronchial epithelia of asthmatic mice. The reduction in cytochrome c oxidase activity was associated with an increase in cytochrome c levels in lung cytosol, a reduction in adenosine triphosphate (ATP) levels in the lung, apoptosis of the airway epithelia, and asthmatic features. All these features were restored with treatments with either dexamethasone or IL-4mAb, indicating the involvement of Th2-mediated inflammation causing such mitochondrial structural changes and dysfunction in asthmatic airway epithelia [67], which also indicates the possible causative role of IL-4 in causing mitochondrial dysfunction in asthma.

While it is known that both IL-4 and IL-13 promote IgE class switching through STAT-6, they also induce an enzyme: 12/15-lipoxygenase (12/15-LOX, also called 15-lipooxygenase in humans) [68]. Even though the role of 5-lipoxygenase (5-LOX) is well established in asthma pathogenesis, the detailed role of 12/15-lipoxygenase was demonstrated by our lab. 12/15-LOX is one of the enzymes responsible for cellular suicide via the programmed disappearance of mitochondria and other cellular organelles from the reticulocytes and for immature fibroblasts converting into red blood cells and mature fibroblasts, respectively [69,70,71]. This is essential for the uninterrupted functions of red blood cells and mature fibroblasts, as the presence of organelles in these cells disturbs their functions, such as effective gas exchange and clear vision, respectively. Though it was known earlier that 12/15-LOX was increased in asthmatic lungs [72,73], its role in mitochondrial dysfunction was not known. The mere overexpression of 12/15-LOX in naïve mice causes mitochondrial dysfunction, along with the development of asthma-like features, indicating the pathogenetic role of 12/15-LOX [74]. Interestingly, it was observed that the bronchial epithelial cells of both control and asthmatic mice did not express 12/15-LOX. However, inflammatory cells, particularly macrophages, had shown significant expressions of 12/15-LOX in asthmatic lungs, and these macrophages may have released the linoleic and arachidonic acid metabolites of 12/15-LOX, like 13-S-HODE (13-Hydroxyoctadecadienoic acid) and 12-S-HETE (12-S-hydroxyeicosatetraenoic acid), respectively [74,75]. The intranasal administration of 13-S-HODE to the naïve mice alone caused mitochondrial degradation along with the development of breathing difficulties, even without allergen exposure. Detailed studies have shown that 13-HODE increases calcium levels in the bronchial epithelia through the transient receptor potential cation channel subfamily V member 1 (TRPV1) calcium channel, which then disrupts calcium homeostasis. When TRPV1 was inhibited in mice, damage to epithelia was reduced, along with the restoration of mitochondrial function. Thus, TRPV1 is a downstream target in the linoleic acid-driven pathway in asthma [75]. Further, 12/15-LOX inhibitors, like esculetin, coumarin-based antioxidants, and baicalein, and also antioxidants, like vitamin E, have been shown to attenuate airway epithelial injury along with the attenuation of asthmatic features [76,77,78]. More efforts are needed to identify 15-LOX modulators as epithelial protective agents in asthma therapy. All these indicate the possibility that mitochondrial dysfunction (observed in the airway epithelium of asthmatic conditions) could be mediated by infiltrated immune cells and cytokines, like IL-4 (released by them). In addition to these findings, the expression of ETC enzymes was also reduced in the blood cells of asthmatic patients even though there were higher mitochondrial copy numbers in these cells, indicating the possible compensatory response for reduced ETC enzymes or more oxidative stress [79].

## 5. Governing/Immune Role of Airway Epithelium in Allergic Airway Inflammation

So far, we have discussed how airway epithelium has been projected as a helpless victim of the immune cells. Although the innate immune response of airway epithelium is well known, it is only in the last decade that the literature has started showing the possible governing role of airway epithelium in maintaining lung homeostasis. 

### 5.1. Less Dominant Role of Inflammation in Causing Epithelial Barrier Dysfunction

The Th2 immune response has been demonstrated as the causative factor for inciting the loss of epithelial layer integrity. However, numerous pieces of literature also suggest the possibility of inflammation-independent epithelial cell dysfunction. Numerous asthma susceptibility genes, for instance, IL33, IL1RL1, MUC5AC, TSLP, CDHR3, and KIF3A, are expressed in the airway epithelium. This highlights the significance of the airway epithelium in the development of asthma [12]. Anomalies in the epithelial barrier due to disruptions in the tight and adherens junctions have been proclaimed to be involved in allergen sensitization and asthma advancement [54]. All these indicate that asthmatic people might have a compromised and dysfunctional epithelial barrier. Genome-wide association studies in asthmatics have shown the association of various genes with asthma susceptibility and how these genes are also expressed in the airway epithelium [80]. In addition, various genes that are crucial in epithelial barrier homeostasis, like protocadherin 1 (PCDH1) and Cadherin-related family member 3 (CHHR3), have also been shown to be associated with asthma genetic studies [80,81,82]. Asthmatic children suffering from severe wheezing have reduced expression of CDHR3 (Cadherin-related family member 3) [83]. In early asthma cases, there was an increase in the expression of ORMLD3, which, in turn, reduced the expression of other epithelial proteins, like claudin 17 and E-cadherin, which gave rise to a compromised epithelial barrier [45]. 

In addition to the inherent barrier defects seen in asthma patients, allergens directly affect the lung epithelial barrier, the first layer of defense in them. Allergens, such as house dust mites (HDMs), pollens, cockroach extracts, and fungi, produces or contain proteases and hence disturb the epithelial barrier, causing increased sensitization [84]. Inhaled allergens or proteases can cause epithelial cells to recognize and react by activating a variety of PRRs, including TLR and PAR. NF-κB activation is stirred up by these activated receptor signals. This, in turn, causes transcriptional activation of myriads’ pro-inflammatory genes, including cytokines and chemokines. As the role of NOD-like receptors in allergic inflammation is complex and context-dependent [85], the role of innate immune receptors in allergic inflammation is complex. On the one hand, prolonged stimulation with low doses of innate immunity receptor agonists before sensitization reduces the severity of the allergic process, and on the other hand, stimulation of innate immunity receptors, together with the action of an allergen, increases allergic inflammation. In any event, the aforesaid evidence indicates that epithelial barrier dysfunction can be an inflammation-independent phenomenon. 

### 5.2. Less Dominant Role of Inflammation in Causing Mitochondrial Dysfunction in Airway Epithelia

Similar to epithelial barrier dysfunction, mitochondrial dysfunction in asthmatic airway epithelia can additionally be unaided by inflammation. Investigations and research have already determined the possibility of inflammation and oxidative stress-induced mitochondrial dysfunction in asthmatic airway epithelia. When there was a forced reduction in the expression of certain ETC enzymes in airway epithelium, allergic airway inflammation features got developed [86]. This indicates the possibility of inflammation-independent mitochondrial dysfunction in asthma pathogenesis. 

In mice studies, the dysfunctional mitochondria from stressed airway epithelium were replaced with the mesenchymal stem cells (MSCs) by overexpressing calcium-sensitive mitochondrial GTPase Miro1. It was observed that there is attenuation in the features of allergic airway inflammation. To such a degree, asthma features in mice got reduced by mesenchymal stem cell-mediated replacement of dysfunctional mitochondria from airway epithelium. Such evidence indicates not only the therapeutic importance of healthy mitochondria but also the determining role of airway epithelium in asthma pathogenesis [87]. Although most of these studies were performed in mice models, various human studies have likewise focused on the importance of mitochondrial genomics and dysfunctional mitochondria in asthma phenotype. Similar to asthma, even in acute lung injury, it has been shown that MSCs can restore the alveolar epithelial function through mitochondrial transfer [88]. In this way, recovery of epithelial health is sufficient to reduce the inflammation both in asthma and acute lung injury. This suggests not only the therapeutic roles of mitochondria but also the governing role of the epithelium in modulating pulmonary inflammation. 

### 5.3. Airway Epithelium Induces ILC2-Mediated Type 2 Immune Response through Alarmins (Figure 3)

So far, the conventional rationale about the airway epithelium is of a ‘victim’ that often comes under recurrent stress from the inflammatory system in asthma. Albeit, over the last decade, mounting evidence has shifted the paradigm for the role of airway epithelium towards a more upstream modulator in airway inflammation. Several PRRs, including TLR2, TLR4, NOD1 [89,90], and PAR1-4, are expressed by airway epithelial cells [91,92]. Upon stimulation, they activate various signaling pathways that lead to the enlistment of immune cells and also the Th2 immune response [93]. One major group of cytokines which is recognized as important in this epithelium-driven immune response is called ‘alarmins’ [94]. ‘Alarmins’, specifically thymic stromal lymphopoietin (TSLP), interleukin-33, and interleukin-25, are released by airway epithelium upon cellular stress or damage caused by an allergen, pathogen, or pollutant exposure ultimately skews the immune response toward type 2 [95,96]. The concept of alarmins was introduced by Joost J Oppenheim. He used this umbrella term to describe a group of host proteins that are released upon cell damage or pathogen challenge, which recruit and activate both innate and adaptive immunity and galvanize the whole immune response through ‘early signals’ [94]. Although airway epithelium is considered the major source of alarmins but innate, adaptive, and other structural cells can also secrete alarmins [97]. 

**Figure 3 diagnostics-13-00808-f003:**
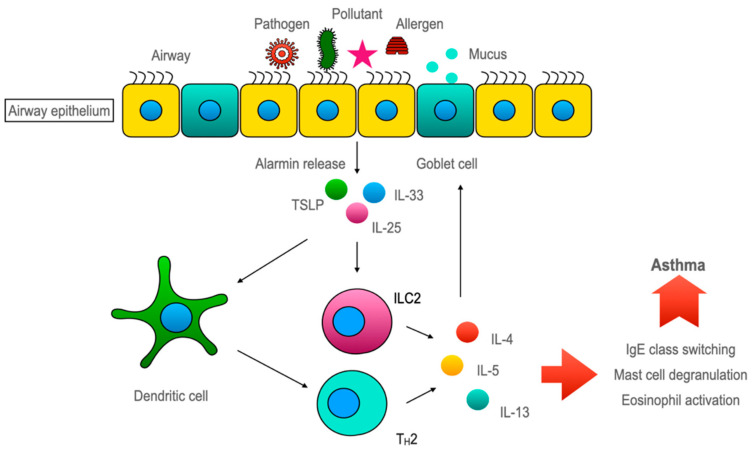
Airway epithelia regulating epi-immune response upon allergen/pathogen exposure. Pathogens or allergens disrupt the airway epithelia. Disrupted epithelia releases alarmins (IL-25, IL-33, and TSLP). Alarmins activate dendritic cells for Th2 polarization of the immune response. On the other hand, alarmins can directly activate ILC2 cells to secret IL-4, IL-5, and IL-13 cytokines. The dysregulation of this pathway leads to asthma pathogenicity.

#### 5.3.1. TSLP

Thymic stromal lymphopoietin, which was initially identified as a member of the IL-2 family, is important for pre-B cell maturation [98]. A variety of stimuli can cause TSLP release from airway epithelium, such as respiratory syncytial virus infection [99], exposure to pro-inflammatory and T_2_ cytokine [92], or activation of PRR by allergen exposure [100]. TSLP is distantly related to IL-7, which is evident by its use of a heterodimeric receptor complex composed of TSLPR and IL-7Rα. TSLP receptor complex is expressed in a number of hematopoietic cells [98] and smooth muscle cells. TSLP activates dendritic cells, which polarize naïve CD4 + T cells to the Th2 phenotype. TSLP can also activate mast cells, eosinophils, and ILC2 cells, all are known for asthma pathogenesis [98]. 

#### 5.3.2. IL-33

IL-33 is a member of the IL-1 cytokine family expressed constitutively by endothelial cells, fibroblasts, airway smooth muscle cells, and epithelial cells [98]. It is synthesized as a precursor with nuclear localization signal predominantly bound to chromatin. With DNA binding ability and cytokine activity, IL-33 can regulate transcription when in the nucleus but also act as an inflammatory cytokine when released from the cell. The receptor for IL-33 is heteromeric made up of suppression of tumorigenicity 2 (ST2) and IL-1 receptor accessory proteins (IL1RAcP) [98]. ST2 encoded by the IL1RL1 gene has two isoforms: ST2L (transmembrane) and sST2(soluble isoform). sST2 isoform acts as a decoy receptor and negatively regulates IL-33 activity. ST2L transmembrane receptors are found on a lot of hematopoietic cells, such as mast cells, macrophages, eosinophils, basophils, Th2 cells, and ILC2 [98]. Inhibition of IL-33 and ST2L interaction ameliorated allergic asthma in a murine model [101]. An increase in IL-33 in airway epithelium and BAL fluid correlated with the severity of asthma patients [102].

#### 5.3.3. IL-25

IL-25 is a member of IL-17 family of cytokines. But unlike other members (IL-17A or IL-17F), which are involved in neutrophilic inflammation and T1 immunity, IL-25 promotes the T2 immune response [98]. Initially, it was thought that eosinophil and basophil were the primary sources of IL-25, but recent shreds of evidence indicate that endothelial cells, specialized chemosensory epithelial cells, activated Th2 cells, macrophages, and fibroblasts all secrete IL-25 [98]. IL-25 is stored as a preformed cytokine in the cytoplasm and is released after exposure to protease-containing allergens, like house dust mites. The IL-25 receptor, a heterodimer of IL-17RA/IL-17RB, is expressed on memory Th2, ILC2, dendritic cells, eosinophils, mast cells, and endothelium. The binding of IL-25 to its receptor activates both adaptive and innate immune responses associated with Th2-type inflammation. Blocking IL-25 improved the asthma condition in a murine model [103]. An increase in IL-25 is associated with greater airway eosinophilia, higher MUC5B expression, subepithelial fibrosis, and higher IgE levels, which was first identified in a subgroup of “IL-25 high” patients by Cheng et al. [104]. 

#### 5.3.4. Discovery of ILC2 and Its Role in Asthma

Fort et al. reported that the intranasal administration of IL-25 in mice increased the type 2 immune response, elevated IgE levels, eosinophilia, mucus production, and epithelial hyperplasia in both normal mice [105] and RAG-deficient mice which lacked both B and T cells [106]. Later on, another group of researchers noticed a cell type that was Lin^−^ cKit^+^ FcεR1^−^ that produces IL-4, IL-5, and IL-13 after IL-25 stimulation [107]. Then, in 2010, three separate groups of researchers ultimately identified innate lymphoid cell type 2, which produces IL-4, IL-5, and IL-13 after stimulation with both IL-25 and IL-33 [108]. As discovered by Kondo et al., the ILC2-mediated allergic response is independent of adaptive immunity. Those RAG-deficient mice lacking functional B and T cells also developed allergic responses against protease-containing allergens, such as HDM. The adoptive transfer of ILC2 cells restored the hypersensitivity in RAG-deficient ILC2-depleted mice against allergens [109]. This evidence indicates that ILC2 could be a crucial player in airway hyperresponsiveness. The existing information suggests that ILC2 and Th2 cells might have the same/equal level of contribution in asthma [110]. Besides these, it is also important for homeostatic tissue repair after a brief epithelial injury, such as with an influenza infection [111]. As the airway constantly comes into direct contact with environmental particles, ILC2 promotes the Th2 response, which is a milder form of inflammation and reduces the damage, but when this is dysregulated, it results in hypersensitivity [112].

#### 5.3.5. Determinant Role of Airway Epithelium in Mounting the Type of Immune Response

Thymocyte development provided the first evidence of epithelium-guiding immune cell maturation. As the concept of peripheral antigen presentation is dominated by DC, the role of epithelia is mostly ignored. However, new findings suggest that the airway epithelium might be at the fulcrum of the inflammatory response, in which they come into direct contact with foreign particles. If epithelium comes under any kind of stress, it releases immune modulators. For example, any wound or abrasion can cause the upregulation of the S100 group of calcium-binding proteins that help to maintain a hyperplastic environment [113], while it also has chemoattractant properties that activate RAGE/TLR-4-dependent pathways in myeloid cells [114]. Similar to immune cells, NF-κB also plays a central role in this epithelium-supervised epi-immune response [115]. Depending on the stimuli, airway epithelium can modulate immune responses. If the epithelial damage is not severe, then TSLP, IL-33, and IL-25 dominate the secreted products, which skews the immune response towards milder type 2. However, upon severe damage, epithelium releases IL-1α, Type-1 INF, and TNF- α, which promote a strong type-1 response. If dysregulated type 2 responses may cause asthma, atopic dermatitis, and ulcerative colitis while dysregulated type 1 responses may result into COPD, Crohn’s disease, Psoriasis, etc. [116].

#### 5.3.6. Airway Epithelium as a Target for Innovative Treatments in Asthma

While ILC2 cells secrete good amounts of IL-4, IL-5, and IL-13 after stimulation with alarmins, the IL-33-mediated activation of ILC2 is resistant to glucocorticoid administration [117]. This evidence indicates epithelia might be involved in a steroid-resistant phenotype. Most asthma patients respond to glucocorticoid and β_2_ agonists therapy except a few (5–10%) [118]. Other than Th2/type 2 steroid-resistant asthma, neutrophilic-steroid-resistant asthma is also prominent, with an increase in the activity of three cytokines: IL-17, IL-8, and TNF-α [119]. Recent studies indicated that a subclass of innate lymphoid cells, ILC1 and ILC3, increase when LPS was administered to neutrophilic asthmatic mice. Interestingly, the eosinophilic ILC2 levels did not get altered. These ILC3 cells can potentiate the release of IL-17 and IL-22 in the airways. This is coupled with diminished epithelial barrier integrity, owing to the reduced expression level of tight junction proteins like occludin, zonula occludens, and claudin [120].

Thus, the experimental evidence indicates that the concept of the canonical activation of the immune system may not be the complete picture. Airway epithelium is not an innocent victim but is rather the dominant decision-maker regulating the immune responses. As asthma is a heterogenous disease, all patients do not respond to glucocorticoids. Identifying the subtypes that respond to different therapies could provide clinicians with more options. The alarmins released after epithelial stress act as an upstream regulator of asthma pathogenesis. Targeting upstream regulators has a more profound effect than any downstream mediator. Therefore, next-generation therapies for asthma treatment could be targeted against these alarmins. Tezepelumab is a monoclonal antibody against TSLP [121]. It has shown promising results in clinical trials and was recently approved by the US FDA for the treatment of severe asthma [122]. Another anti-TSLP monoclonal antibody is also undergoing clinical trials. Anti-IL-33 therapy is also under consideration, as the several monoclonal antibodies that target IL-33 are in different stages of clinical trials [123]. T_H_2/type 2 steroid resistance is the result of the alarmin-mediated activation of ILC2 [117]. Therefore, strategies that target alarmins or their release could be a good approach for asthma management. 

## 6. Conclusions

When the treatment or control of asthma was insufficient, there was a need to modify the existing concept of asthma pathogenesis. Thus, the concept of asthma pathogenesis changed periodically from a neuropsychological disease to airway inflammation. Parallel to this conceptual change, the treatment strategy also changed from “calming mind” to anti-inflammatory drugs. For a long time, it has been believed that the Th2 immune response is the main driver of the entire pathophysiological features of asthma. When asthma was considered a Th2-dominant inflammatory disease, it was believed that almost everything was solved in asthma pathogenesis. However, this Th2-dominant hypothesis could not explain the poor correlation between airway inflammation and airway remodeling, severe asthma endotypes [Th2-high (eosinophilic) and Th2-low (non-eosinophilic)], therapy resistance in a certain percentage of asthmatics, etc. This indicated a need to change the concept used in asthma pathogenesis. In this context, it was always believed that the airway epithelium was a helpless victim of the immune cells. Although this was demonstrated to be partially true, it seems this is not the entire story. In this context, the literature in the last decade has emphasized the importance of airway epithelium in asthma pathogenesis. The airway epithelium has dual roles in healthy lung homeostasis and asthmatic lungs. The airway epithelium maintains lung homeostasis against environmental irritants, with its armamentaria, including anatomical barriers, chemical barriers, chemosensory apparatus, like pulmonary neuroendocrine cells, a detoxification system for inhaled xenobiotics, etc. However, if the external allergens load is severe and beyond the capacity of the airway epithelium, with or without the inherent epithelial barrier dysfunction in asthmatics, the airway epithelium starts secreting special biomolecules called alarmins that induce the ILC2-mediated type 2 immune response. This role is pathogenetic, as these alarmins tend to amplify the inflammatory response in contrast to the earlier homeostatic role. However, the exact role of these alarmins in asthma pathogenesis is yet to be investigated. The inflammatory mediators released from Th2 cells induce oxidative stress and also epithelial cell death. If the external irritants are severe, as is the case with cigarette smoke, the airway epithelium also releases IL-17A and IL-8, which are crucial in neutrophilic asthma. Current asthma pathogenesis knowledge is mostly based on the concept of “inflammation-induced epithelial injury”. So, it is obvious that therapeutic strategies have not focused on improving epithelial functioning in asthmatics, holding the belief that airway inflammation is the upstream event when an epithelial injury is caused. Thus, we believe that an epithelial-driven concept in asthma pathogenesis could fill most of the gaps in the current knowledge of asthma and also leads to the future possible incorporation of epithelial-protective agents to enhance the strength of the epithelial barrier and increase the fighting capacity of the airway epithelium against exogenous irritants/allergens and, thus, reduce asthma incidence and severity, with better asthma control. 

## Data Availability

All the data are available in the manuscript.

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
