# Peer review of "Airway Epithelium: A Neglected but Crucial Cell Type in Asthma Pathobiology"

_diagnostics, 2023, doi:10.3390/diagnostics13040808_

Round 1

Reviewer 1 Report

Manuscript ID: Diagnostics-2200967

Type: Review

Title: „ Airway epithelium: A neglected but crucial cell in asthma 2 pathobiology.”

The article is interesting, the idea of the important role of the epithelium in the development of asthma and the importance of search of therapeutics that act on the increase the strength of the epithelial barrier and the ability of the respiratory epithelium to combat exogenous irritants/allergens is neglected. This novel approach to understand asthma pathology requires much attention.  However to be published article needs to be improved. I have a lot of comments. In general, many citations are missing from the text.

1. Section 2 and Fig. 1, b-agonist change to greek, b-agonists.

2. Line 250-251. Please provide some reference to this information.

3. Section 3.4 is based on only one reference, please give some more.

4. Line 275. Please after ”them” add some reference.

5. Line 285. What about references to COPD, and small cell lung cancer?

6. Line 291, Please give some reference to this statement.

7. Unify writing “Clara” or “clara” cells

8. Line 326. Please after ”barrier dysfunction” add some reference.

9. line 330. Add some short information and reference on the low weight molecular agents and its ability to trigger asthma

10. Line 352. Remove “while” it will be more reader friendly

11. Line 358. Explain shortcut” HBEC”.

12. Line 369. Please after ”barrier dysfunction” add some reference.

13. Line 373. Remove Pelaia et al., 2019.

14. Line 381. Please remove “further”, since it suggests that you performed another study, and it is still the same reference 54.

15. Line 394, After [54] add “which”.

16. Add shortcut expansion: 5-LOX, 14/15LOX

17. Line 412,. Please provide some reference to this information.

18. Line 446. Please provide some reference to GWAS study.

19. Line 490, shortcut PRR was earlier explained in the text.

20. Line 494. It should be is called “alarmins”

21. Line 514. Please add “IL-33 is a member”, otherwise this sentence doesn’t make much sense.

22. Line 521, It should be “acts”

23. Line 538. Please provide some reference to this murine study.

24. Line 560. Please provide some reference after “hypersensitivity”.

Author Response

Reply to Comments of Reviewers (Manuscript ID: Diagnostics-2200967)

Reviewer 1:

General comments:

The article is interesting, the idea of the important role of the epithelium in the development of asthma and the importance of search of therapeutics that act on the increase the strength of the epithelial barrier and the ability of the respiratory epithelium to combat exogenous irritants/allergens is neglected. This novel approach to understand asthma pathology requires much attention.  However to be published article needs to be improved. I have a lot of comments. In general, many citations are missing from the text.

Reply to general comments: Thanks for the appreciation. We agree with you regarding the missing citations. We have added more references for more clarification. 

Comment 1. Section 2 and Fig. 1, b-agonist change to greek, b-agonists.

Reply 1: We have the changed the same in Fig.1.

Comment 2. Line 250-251. Please provide some reference to this information.

Reply 2: We have added the references.

Comment 3. Section 3.4 is based on only one reference, please give some more.

Reply 3: We have added more references.

Comment 4. Line 275. Please after ”them” add some reference.

Reply 4: We have added two references.

Comment 5. Line 285. What about references to COPD, and small cell lung cancer?

Reply 5: We have added few references.

Comment 6. Line 291, Please give some reference to this statement.

Reply 6: We have added few references.

Comment 7. Unify writing “Clara” or “clara” cells

Reply 7: Done

Comment 8. Line 326. Please after ”barrier dysfunction” add some reference.

Reply 8: We have added few references.

Comment 9. line 330. Add some short information and reference on the low weight molecular agents and its ability to trigger asthma

Reply 9: We have added short information and references.

Comment 10. Line 352. Remove “while” it will be more reader friendly

Reply 10: We have removed the same.

Comment 11. Line 358. Explain shortcut” HBEC”.

Reply 11: We have expanded.

Comment 12. Line 369. Please after ”barrier dysfunction” add some reference.

Reply 12: We have added few references.

Comment 13. Line 373. Remove Pelaia et al., 2019.

Reply 13: We have removed the same.

Comment 14. Line 381. Please remove “further”, since it suggests that you performed another study, and it is still the same reference 54.

Reply 14: We have removed the same.

Comment 15. Line 394, After [54] add “which”.

Reply 15: We have added the same.

Comment 16. Add shortcut expansion: 5-LOX, 14/15LOX

Reply 16: We have expanded them.

Comment 17. Line 412,. Please provide some reference to this information.

Reply 17: We have added reference.

Comment 18. Line 446. Please provide some reference to GWAS study.

Reply 18: We have added reference.

Comment 19. Line 490, shortcut PRR was earlier explained in the text.

Reply 19: Yes. It was. So we have given only short form here.

Comment 20. Line 494. It should be is called “alarmins”

Reply 20: Typo is corrected.

Comment 21. Line 514. Please add “IL-33 is a member”, otherwise this sentence doesn’t make much sense.

Reply 21: We have modified the same to improve clarification as suggested.

Comment 22. Line 521, It should be “acts”

Reply 22: Typo is corrected.

Comment 23. Line 538. Please provide some reference to this murine study.

Reply 23: We have added reference.

Comment 24. Line 560. Please provide some reference after “hypersensitivity”.

Reply 24: We have added reference.

Reviewer 2 Report

The article is deal with the integrity of airway epithelium during asthma. The topic discussed is very important for the treatment and prevention of asthma.

I would like to make a few comments:

1)      Information about the types of asthma is incomplete, article need to be discussed and cited:

Sze E, Bhalla A, Nair P. Mechanisms and therapeutic strategies for non‐T2 asthma. Allergy. 2020;75:311–325. https ://doi.org/10.1111/all.13985

2)      It is necessary to discuss not only the influence of TLR receptors on the pathogenesis of asthma, but also NOD2 receptors of innate immunity. Cite the article:

Guryanova, S.V.; Gigani, O.B.; Gudima, G.O.; Kataeva, A.M.; Kolesnikova, N.V. Dual Effect of Low Molecular Weight Bioregulators of Bacterial Origin in Experimental Model of Asthma. Life 2022, 12, 192. https://doi.org/10.3390/life12020192

and discuss the complexity of innate immune receptor`s regulation of allergic inflammation. In particular, prolonged stimulation with low doses of innate immunity receptor agonists before sensitization, reduces the severity of the allergic process. In the case of stimulation of innate immunity receptors together with the action of an allergen, allergic inflammation increases.

3)      Also indicate that coronaviruses destroy the ciliated epithelium and violate its integrity:

Adivitiya; Kaushik, M.S.; Chakraborty, S.; Veleri, S.; Kateriya, S. Mucociliary Respiratory Epithelium Integrity in Molecular Defense and Susceptibility to Pulmonary Viral Infections. Biology 2021, 10, 95. https://doi.org/10.3390/biology10020095

4)      Discuss and cite the article:

Jonckheere A-C, Seys SF, Steelant B, Decaesteker T, Dekoster K, Cremer J, Dilissen E, Schols D, Iwakura Y, Vande Velde G, Breynaert C, Schrijvers R, Vanoirbeek J, Ceuppens JL, Dupont LJ and Bullens DMA (2022) Innate Lymphoid Cells Are Required to Induce Airway Hyperreactivity in a Murine Neutrophilic Asthma Model. Front. Immunol. 13:849155. doi: 10.3389/fimmu.2022.849155

Author Response

Reply to Comments of Reviewers (Manuscript ID: Diagnostics-2200967)

Reviewer 2:

General comments: The article is deal with the integrity of airway epithelium during asthma. The topic discussed is very important for the treatment and prevention of asthma.

Reply to general comments: Thanks for the appreciation. We agree with you regarding the missing citations. We have added more references for more clarification. 

I would like to make a few comments:

     Comment 1)      Information about the types of asthma is incomplete, article need to be discussed and cited:

Sze E, Bhalla A, Nair P. Mechanisms and therapeutic strategies for non‐T2 asthma. Allergy. 2020;75:311–325. https ://doi.org/10.1111/all.13985

Reply 1: We have discussed and cited the mentioned article.

     Comment 2)      It is necessary to discuss not only the influence of TLR receptors on the pathogenesis of asthma, but also NOD2 receptors of innate immunity. Cite the article:

Guryanova, S.V.; Gigani, O.B.; Gudima, G.O.; Kataeva, A.M.; Kolesnikova, N.V. Dual Effect of Low Molecular Weight Bioregulators of Bacterial Origin in Experimental Model of Asthma. Life 2022, 12, 192. https://doi.org/10.3390/life12020192

and discuss the complexity of innate immune receptor`s regulation of allergic inflammation. In particular, prolonged stimulation with low doses of innate immunity receptor agonists before sensitization, reduces the severity of the allergic process. In the case of stimulation of innate immunity receptors together with the action of an allergen, allergic inflammation increases.

Reply 2: We have discussed and cited the mentioned article.

     Comment 3)      Also indicate that coronaviruses destroy the ciliated epithelium and violate its integrity:

Adivitiya; Kaushik, M.S.; Chakraborty, S.; Veleri, S.; Kateriya, S. Mucociliary Respiratory Epithelium Integrity in Molecular Defense and Susceptibility to Pulmonary Viral Infections. Biology 2021, 10, 95. https://doi.org/10.3390/biology10020095

 Reply 2: We have discussed and cited the mentioned article.

      Comment  4)      Discuss and cite the article:

Jonckheere A-C, Seys SF, Steelant B, Decaesteker T, Dekoster K, Cremer J, Dilissen E, Schols D, Iwakura Y, Vande Velde G, Breynaert C, Schrijvers R, Vanoirbeek J, Ceuppens JL, Dupont LJ and Bullens DMA (2022) Innate Lymphoid Cells Are Required to Induce Airway Hyperreactivity in a Murine Neutrophilic Asthma Model. Front. Immunol. 13:849155. doi: 10.3389/fimmu.2022.849155

Reply 4: We have discussed and cited the mentioned article.

Round 2

Reviewer 1 Report

The work has been significantly revised, thus I recommend it for publication as it stands.